# Variations in Behavioral and Physiological Traits in Yearling Tibetan Sheep (*Ovis aries*)

**DOI:** 10.3390/ani11061676

**Published:** 2021-06-04

**Authors:** Yibo Yu, Yun Wang, Liang Zhong, Hongjuan Zhu, Jiapeng Qu

**Affiliations:** 1Key laboratory of Adaptation and Evolution of Plateau Biota, Northwest Institute of Plateau Biology, Chinese Academy of Sciences, Xining 810008, China; yuyibo@nwipb.cas.cn (Y.Y.); zhongliang@nwipb.cas.cn (L.Z.); zhuhongjuan@nwipb.cas.cn (H.Z.); 2Research Center for Environment Protection and Water and Soil Conservation, China Academy of Transportation Sciences, Beijing 100029, China; wangyun80314@163.com; 3Qinghai Province Key Laboratory of Animal Ecological Genomics, Xining 810008, China; 4University of Chinese Academy of Sciences, Beijing 100049, China

**Keywords:** behavior, Tibetan sheep, ontogeny, temperament, physiology

## Abstract

**Simple Summary:**

Tibetan sheep (*Ovis aries*) are raised in the wild by shepherds on the Tibetan Plateau and domesticated for a long period. Studying variations in the behavioral and physiological traits of sheep during the growing season is important for welfare and productivity. In this study, the behavior and physiology of a Tibetan sheep cohort were repeatedly evaluated in 2-, 5-, and 11-month-old sheep. The mean level of the risk-taking variable increased, while that of the vocalizations variable decreased. The exploration variable was stable over ontogeny. The novelty decreased, and the heart rate increased from 2 to 5 months, but both stabilized at 11 months. The fecal cortisol concentration (CORT) variable was stable at 2 and 5 months but decreased at 11 months of age. Stable correlations were reported for 2- and 5-month-olds and for behavioral variables and heart rate. However, some correlations emerged only at 11 months, whereas others disappeared over ontogeny. Moreover, CORT was independent of temperament and heart rate during the entire period.

**Abstract:**

Temperament is a consistent behavioral difference among individuals over time or in different contexts. A comprehensive understanding of temperament and complex behavioral interactions enhances knowledge on animal evolution, welfare, and productivity. However, reports on the development of behavioral consistency over ontogeny are vague. Here, we tested the ontogeny of the temperament and physiological traits of Tibetan sheep (*Ovis aries*) in three crucial age stages. The mean level of the risk-taking variable increased, while that of the vocalizations variable decreased. The exploration variable was stable over ontogeny. The novelty decreased and the heart rate increased from the juvenile to the adolescent stage but stabilized at the adult stage. The fecal cortisol concentration (CORT) variable was stable at the juvenile and adolescent stages but decreased at the adult stage. Stable correlations were reported for the juvenile and adolescent stages and for the behavioral variables and heart rate. However, some correlations emerged only after maturation, whereas others disappeared over ontogeny. Moreover, CORT was independent of temperament and heart rate at different ages. These results demonstrate that age affects temperament and physiology and their correlations. Hence, developmental aspects should be incorporated into future temperament studies.

## 1. Introduction

Temperament, also termed personality, is a consistent inter-individual difference in animal behavior and is relatively stable over time and/or contexts [1]. Temperament has been proven to be hereditary to some extent. For instance, the risk-taking and docility of bighorn sheep (*Ovis canadensis*) were highly inherited between ewes and lambs [2]. Temperament is considered an important component of life history strategies because it affects animals’ responses to different situations [2]. For instance, compared with shy individuals, bold Scottish Blackface sheep are insensitive to external interferences and their spatial distribution is scattered, contributing to increased opportunities to graze food and favoring their growth [3]. Exploration and novelty are correlated to dispersal and foraging and may influence the probability of parasitic infection [4,5]. Research on temperament ontogeny contributes to a better understanding of the ecology and evolution of temperament and reveals the mechanisms of developmental constraints and potential life history variations. Moreover, prior experiences during early age can affect temperament in later age; therefore, the developmental perspective can benefit from the investigation of the formation mechanisms of temperament [6].

According to the pace-of-life syndrome hypothesis, temperament is linked to physiology and life history strategies, such as metabolism and digestion [7,8]. Correlations among temperament and physiological traits, termed “coping style” or “behavior syndrome”, can affect all aspects of life history, including sexual selection, reproduction, and growth [9]. Although the behavior syndromes of different taxa have been widely investigated in wild animals and livestock [8,10], studies on the variations and associations of animal temperament over different ontogenic age stages remain rare [11,12].

A primary reason for the absence of experimental studies on temperament development may be the concept of temperament, which is the consistency of behavior over time; moreover, ontogeny studies found that correlations of different traits might disappear at different age stages, thereby disproving the concept of temperament [1]. The tested individuals should be measured across different ontogenetic stages to study the ontogeny of temperament, but this approach is difficult to achieve in the field [13]. Changes in the crucial stages of development may influence animal temperament, even over a short time. For instance, the risk-taking or vocalizations of Bighorn sheep may decrease with age [14]. The exploration of 4-week-old weaned mice became stronger than that of 3-week-old mice [15]. However, studies on the long-term ontogenic development of temperament and physiology remain limited. One of the few studies found that the basal cortisol concentrations (CORTs) in cavies decrease with age, and that the correlations between resting metabolic rate and fearlessness are stable over ontogeny; meanwhile, the correlations between CORTs and exploration disappear after maturation [1]. Research on the ontogeny of temperament in livestock (e.g., sows) can also benefit their welfare and productivity [12].

In this study, we examined the temperament and physiology development of yearling Tibetan sheep (*Ovis aries*) in March (2 months old), June (5 months old), and December (11 months old), i.e., from the suckling to the weaning period. Tibetan lambs grow fast in their first year of life and sexually mature at around 1 year of age. The feeding practices and management of sheep in the high-altitude grasslands of the Tibetan Plateau are based on ecological farming practices. Lambs followed ewes for suckling and grazing at 2 months old, were partly separated from the ewes at 5 months old, and were then completely separated from them at 11 months old; thus, the social environments and ontogenic conditions of the lambs were distinct from March to December. The variations in the temperament (risk-taking, vocalizations, exploration, and novelty) and physiological traits (heart rate and CORT) of the sheep and their correlations over ontogeny were tested. The linear mixed-effect model was used to assess the effects of age, sex, and birth weight on the measured traits. Parametric and non-parametric statistical methods were conducted to determine trait variations and correlations at the three different age stages. This study aims to assess (1) whether the temperament and physiological traits of yearling Tibetan sheep are stable over the developmental stages and (2) whether the correlations between temperament and physiological traits are stable at the different age stages.

## 2. Materials and Methods

### 2.1. Ethics Statement

The procedure for feeding and handling the Tibetan sheep followed the guidelines of the Animal Care and Use Committee and complied with the ethical approval standard of the Ethics Committee of the Northwest Institute of Plateau Biology, Chinese Academy of Sciences (NWIPB20170114).

### 2.2. Animal Breeding

The Tibetan sheep investigated in this study were raised at the Haibei Demonstration Zone of Plateau Modern Ecological Animal Husbandry Science and Technology (Qinghai, China). The sheep grazed freely on natural pastures from 06:00 h to 17:00 h during daytime and were driven back to their pen at dusk. At the pen, artificial fodder (Menyuan Yongxing Ecological Agriculture and Animal Husbandry Development Co., Ltd, Tibetan Autonomous Prefecture of Haibei, China) and drinking water were provided ad libitum to the sheep.

The Tibetan sheep cohort (*n* = 89; 43 females and 46 males) were born between 4 and 23 of January 2017. These lambs were marked with commercial plastic ear tags. The body mass of each lamb was measured repeatedly by an electronic platform scale each month from January to July, and also in December, 2017. These measurements included the birth weight. As the lambs were raised in a large flock, some tested lambs were missing at the second and third tests. The behavioral and physiological traits were measured in March, June, and December. During the testing period, the targeted lambs were randomly selected from the large flock, kept in the pen, and fed with fodder and water. All the sheep were tested by Qu J.P., who was trained 5 days before the experiment, to ensure that all the measurements were consistent.

### 2.3. Behavior Assay

All behavioral tests were conducted in outdoor arenas (20 m × 9 m) enclosed in opaque plastic sheets to avoid the interference of the ambient environment. Figure 1 shows the diagram of the test arenas. Two video cameras (Sony FDR-AX700)(Sony (China) Co., Ltd, Shanghai, China) were installed at the roof of the sheep pen to record the testing episodes. The arena layout was unchanged during the entire experiment. The testing period for a set of tests was initialized with the closing of the arena entrance and the opening of the exit. Each variable was recorded or analyzed by the same person to avoid the observer effect. The lambs were captured and tested by randomly selected ear tags. All experiments were conducted from 8:00 a.m. to 13:00 a.m. Behavioral and physiological traits were measured. After each test, fresh feces were immediately collected, placed into 2-milliliter sterile microcentrifuge tubes, and then kept in a −20 °C refrigerator. At the end of each test, the feces and urine in the arena were cleaned and sprayed with 75% alcohol to ensure that the sheep would not be disturbed by the foul odor that usually accompanied the testing. The testing order of the variables was consistently followed (Table 1).

The risk-taking variable was measured in the single test arena (5 m × 3 m) [3]. Each sheep was allowed to adapt to the single test arena for 5 min and then the gate connected to the open field arena was opened. The initiative of each lamb to move from the single test arena to the open field arena was recorded as the emergence latency variable. If a focal lamb did not enter the open field arena within 2 min, then emergence latency was recorded as 120 seconds, and the lamb was gently driven into the open field arena.

The vocalizations and exploration of the Tibetan sheep were measured in the open field arena (5 m × 9 m). The arena ground was marked with gridlines (1 m × 1 m). The gate was closed gently upon entry of the focal sheep into the open field arena. The number of calls and travel across gridlines were recorded by a video camera for 180 seconds [3]. The number of calls, i.e., vocalizations, was used as the indicator of nervousness because it can represent the stress caused by social isolation [16]. The number of gridlines crossed was used as the indicator of exploration.

Following the open field test, the novelty variable was measured in the novel object arena (5 m × 9 m). Different novel objects, i.e., an orange and white parking cone, a wet floor sign, and a yellow man-shaped “danger” sign, were used to explore the lambs’ novelty at different age stages [17]. Each novel object was placed at opposing sides of the entrance, i.e., 2.5 m from the long side and 1.5 m from the short side. The gate was closed gently as soon as the focal sheep entered the novel object arena, and the time in which the sheep first touched the novel object was recorded. The test lasted for 3 min. If the sheep did not touch the novel object within 3 min, then the time was recorded as 180 s. Shorter times measured in the risk-taking or novelty tests indicated that the sheep was bolder or more curious.

### 2.4. Physiological Assay

The body mass, length, height, chest circumference, and heart rates of the sheep were measured after the novel object test. A sheep was manually constrained, and a digital voice recorder (Sony, PCM-D100, Japan) was positioned close to its left thorax to record its heart rate for 30 seconds. The heart rate was then analyzed by Praat software (Vision 5.4.1, http://www.fon.hum.uva.nl/praat/ (accessed on 9 November 2014). We randomly selected 12 pairs of adjacent peaks of the recorded sound wave spectrum. The interval between one peak and its adjacent peak was recorded as the time of one heartbeat. Heart rate was obtained from the mean of 12 heartbeats. Heart rate is widely used as an index to evaluate the sympathetic response under stress in livestock and wildlife [2,7].

Fecal CORT was measured using a sheep cortisol ELISA kit following the operation manual (Shanghai DUMA Biotechnology Co., Ltd, China). The assay sensitivity was 1.0 ng/mL, and the degree of confidence was >99%. The intra- and inter-assay CVs were 11.2% and 8.4%, respectively.

### 2.5. Statistical Analysis

A Shapiro–Wilk test was conducted to detect if the data followed normal distribution. All variables except CORT did not conform to normal distribution. A Univariate Markov chain Monte Carlo mixed model was used to calculate the repeatability of the behavioral and physiological traits. Sex, age, and birth body mass were set as the fixed effects. Individual identity and test order were used as random factors. A Bayesian-based statistical method was adopted using the R package MCMCglmm and run for 300,500 iterations with a thinning interval of 300 and a burn-in of 500. Variations in temperament and physiological traits and their correlations at the population level were tested. One-way analysis of variance (ANOVA) and least significant difference analysis were used to compare variations in CORTs at different stages, while the other trait variables were analyzed using a Wilcoxon signed-rank test. Spearman’s rank correlation test was used to assess behavioral syndrome. Fischer Z-transformation was used to convert the rho values into Z-values. Changes in the behavioral syndrome for different age groups were evaluated. The q-value was obtained by conducting false discovery rate correction using the Benjamini–Hochberg method. Either the Spearman or Pearson correlation test was used to predict the effects of traits in the early stage on traits in the late stage.

All statistical analyses were conducted using R 3.5.1 software (R Development Core Team, Auckland, New Zealand).

## 3. Results

A total of 71, 51, and 41 Tibetan lambs were measured in March, June, and December, with mean body masses of 14.90 ± 0.42, 24.66 ± 0.59, and 29.70 ± 0.60, respectively. The repeatability of the behavioral and physiological traits ranged from 0.12 to 0.68. Novelty had the lowest repeatability at 0.12 (0.06, 0.23), whereas vocalizations had the highest repeatability at 0.68 (0.52, 0.74).

### 3.1. Mean-Level Difference and Prediction

Exploration was stable at the group level, whereas the other traits varied over ontogeny (Table 2). Risk-taking increased and vocalizations decreased over ontogeny (Figure 2a,b). Novelty decreased and heart rate increased until 5 months of age; they remained stable in the late stage (Figure 2d,e). CORT decreased at 11 months old (Figure 2f).

All traits except novelty and heart rate showed some correlations across different ontogenetic stages (Table 2). Vocalizations or exploration in the 2-month-olds predicted the trends in the 5- or 11-month-olds. Risk-taking and CORT in the 5-month-olds predicted the trend in the 11-month-olds.

### 3.2. Correlations between Traits

The correlations between two traits changed over ontogeny (Table 2). From 2 to 5 months of age, the correlations between risk-taking and exploration were consistently negative. The correlations between risk-taking and vocalizations varied from negative to positive. All these correlations disappeared in the 11-month-olds. Correlations between risk-taking and heart rate emerged in the 5- and 11-month-olds. Correlations between risk-taking and novelty only emerged in the 11-month-olds. Correlations among the other traits only appeared in a single ontogenetic stage, e.g., the correlations between exploration and vocalizations, novelty, or heart rate. CORT was not correlated with any of the other traits over the whole ontogeny.

## 4. Discussion

In this study, we followed a cohort of Tibetan sheep from birth to adulthood to test the ontogeny of their behavioral and physiological traits over time. Six behavioral and physiological traits (risk-taking, vocalizations, exploration, novelty, heart rate, and fecal CORT) were measured. The results indicated variability in the average levels of behavioral and physiological traits and their correlations. As the individuals developed, the older sheep were bolder, less neurotic, and novel; their heart rates increased, while CORTs decreased, i.e., individuals became proactive. These results are consistent with the findings of Horback and Parsons [12], who found that the behaviors of juvenile sows can predict that of adults; that is, all behavioral traits between the two stages were positively correlated.

Natural selection—such as environmental challenges experienced early in life—may generate animal temperament and behavioral syndromes and affect variations in animal behavior [7]. The personalities of wild animals may vary with age under harsh natural selection pressure (e.g., predator risk). For instance, wild Bighorn sheep become shyer and more alert with age [14]. Experience under risk of predation over ontogeny increased the behavioral plasticity and activity predictability of tadpoles [18]. When environmental challenges (e.g., predation risk or space competition) were absent, compared with juvenile individuals, the temperament variations and correlations were strengthened in adult Eastern mosquitofish [19]. In captive or laboratory conditions with no or few external challenges, temperament may be stable or variable over ontogeny. For example, the risk-taking and aggression of the social spider *Stegodyphus sarasinorum* emerged in the sub-adult stage; these traits became persistent in the long term until the adult stage [20]. Conversely, correlations among aggression, activity, and shyness were detected in young zebra finches, but these correlations disappeared in the mature stage [13].

Livestock domestication is the result of artificial selection and breeding to decrease the fear of humans and increase docility for ease of management. When livestock live in relatively safe environments, high risk-taking increases the capacity of the animals to acquire more food recourses or higher reproductive fitness [21]. Similar to guinea pigs [1], following growth and development, Tibetan sheep that were familiar with stable environments and human disturbance became bolder. The exploration of Tibetan sheep was stable over ontogeny because sufficient food was usually provided; non- or low-stress environments benefit sheep by stabilizing their exploration and foraging [22]. Consistent with a previous study [1], the novelty of Tibetan sheep decreased with age and became stable at 11 months of age because individual experiences and independent living after weaning may reduce the animals’ novelty [19].

The physiological traits of Tibetan sheep varied over ontogeny. As a typical social animal, the number of calls (vocalizations) of Tibetan sheep reflects emotional stress under an isolated environment. Vocalizations decreased with age, suggesting that Tibetan sheep become less nervous in isolated environments, especially in independent living quarters. Fecal CORT is generally used as an indicator of chronic physiological stress [23]. The high CORT levels in the 2- and 5-month-olds could indicate that these sheep suffered higher external stress, which decreased when they were 11 months old; this finding indicates that sheepherders should devote more attention to the care of their young sheep. Heart rate reflects an animal’s sympathetic system reactivity and acute stress response [7]. The heart rate of the Tibetan sheep increased from 2 to 5 months of age. The increased heart rate and decreased CORTs confirmed that the Tibetan sheep became more proactive over ontogeny and suffered lower stress at 11 months of age [24].

Some behavioral and physiological traits of the Tibetan sheep were correlated with one another, but these correlations changed along with ontogenetic stages. Risk-taking, vocalizations, and exploration were correlated in 2- and 5-month-olds, but these correlations disappeared in 11-month-olds. The correlations between novelty and risk-taking or exploration only emerged in 11-month-olds, consistent with the studies of Guenther et al. [1], who found that the correlations between exploration and risk-taking disappeared in mature cavies. Similarly, Bosco et al. [25] reported a strong correlation between aggressiveness and exploration in mature male spiders. Exploration, risk-taking, and novelty reflect food foraging ability in open environments [26]. Sexually mature sheep need to forage more food resources to meet their high energy requirements, especially during the reproductive season. The combination of exploration, risk-taking and novelty can aid 11-month-old sheep in obtaining fresh forages on the alpine pasture [27].

Correlations between heart rate and risk-taking or exploration only appeared in the 11-month-olds. This finding is consistent with the report of Réale et al. [7], who found that risk-taking and heart rate were positively but weakly correlated in Bighorn sheep. Our results provide further evidence to support the coping style hypothesis that relates temperament to the response of the sympathetic nervous system to stress [28]. Consistent with the study of Qu et al. [29], the non-correlations between CORT and temperament or heart rate confirmed that parasympathetic system reaction is independent of behavior and sympathetic system reaction in Tibetan sheep, further supporting the two-tier model [30].

## 5. Conclusions

An increasing number of studies indicate variations in animal temperament over ontogenic stages, showing that individual behavior is consistent across the developmental process. In the current study, we report that the behavioral and physiological traits of yearling Tibetan sheep are relatively stable over 1 year. Our results suggest that the early behavioral and physiological responses of lambs under environmental and social pressure may predict future responses to similar pressures experienced by adult sheep.

## Figures and Tables

**Figure 1 animals-11-01676-f001:**
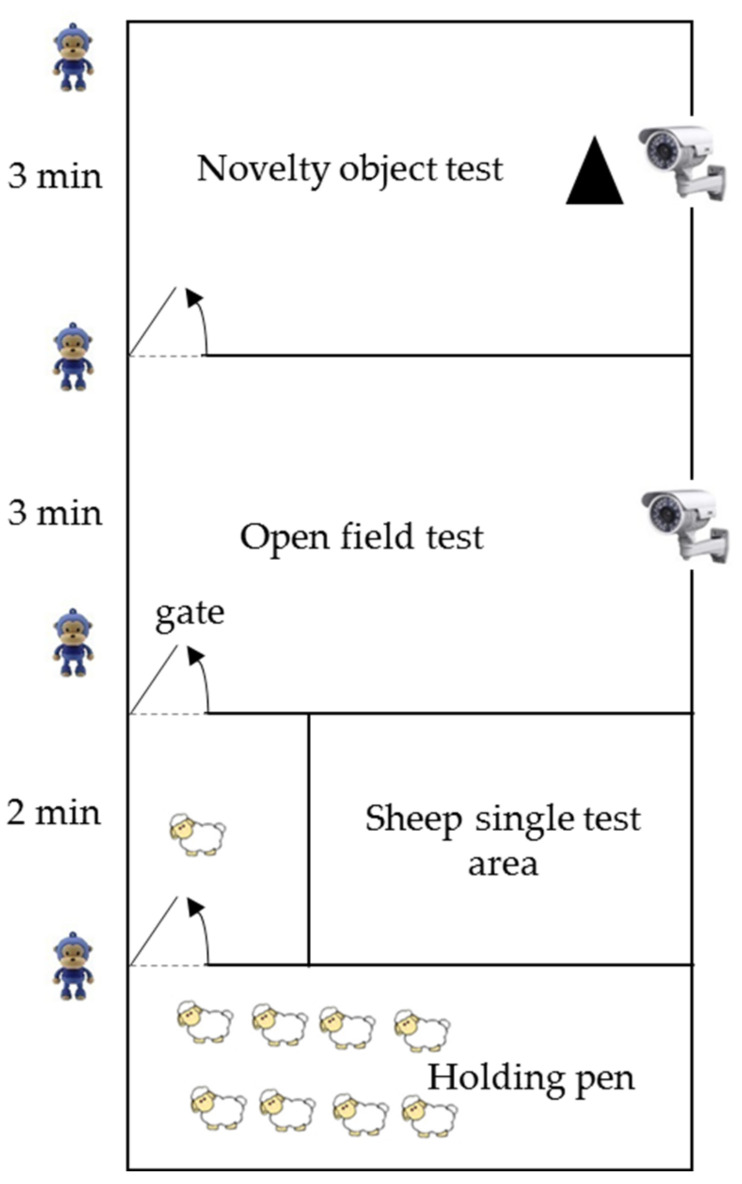
Diagram showing the experimental conditions of the test arena.

**Figure 2 animals-11-01676-f002:**
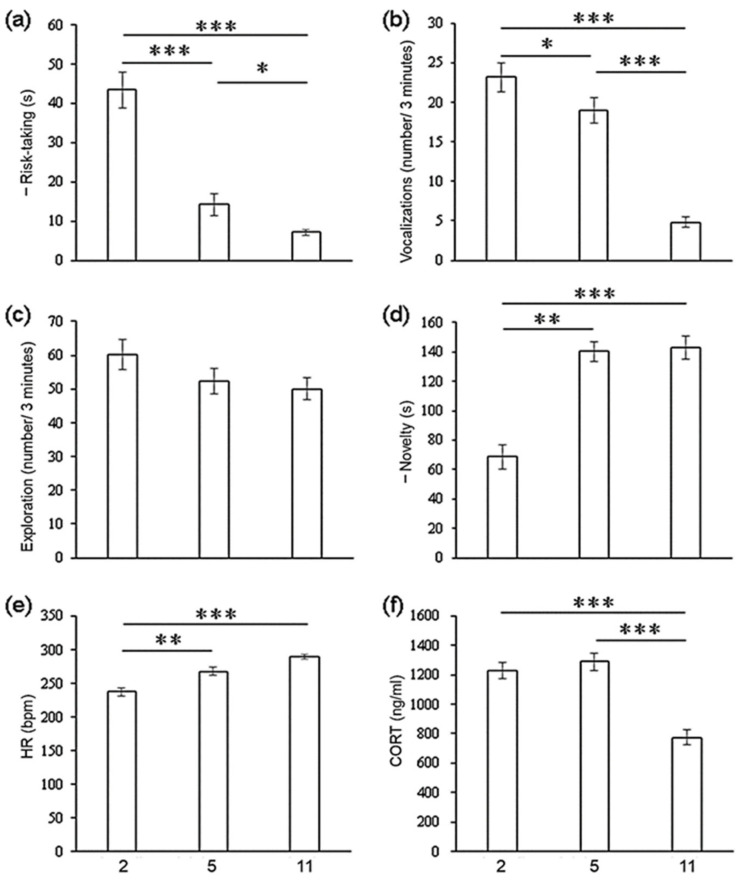
Mean differences in behavioral and physiological traits at three age stages (2, 5, and 11 months old). (**a**) Risk-taking, (**b**) vocalizations, (**c**) exploration, (**d**) novelty, (**e**) HR (heart rate), and (**f**) CORT (fecal cortisol concentration). Mean differences in CORT were analyzed by one-way ANOVA and the LSD method (LSD0.05); differences in the other variables were tested with a non-parametric Wilcoxon signed-rank test. Error bars indicate standard error. * *p* < 0.05, ** *p* < 0.01, *** *p* < 0.001. Note: “—” indicates higher risk-taking and novelty and shorter time.

**Table 1 animals-11-01676-t001:** Description of the behavioral and physiological traits of Tibetan sheep (*Ovis aries*).

Variable	Interpretation of Each Variable
Risk-taking	Emergence latency, the time from the single test area to the open field (s)
Vocalizations	Number of calls in the open field (number/3 min)
Exploration	Divided the surface of the open field into a grid with equal-sized squares (1 m × 1 m) and counted the number of gridlines passed by the Tibetan sheep as exploration. The Tibetan sheep with a body size covering more than half a square plus one or less than half a square did not count (number/3 min)
Novelty	Time before first touching a novel subject (s)
Heart rate	Heart rate under handling (bpm)
CORT	Cortisol concentration in feces (ng/mL)

**Table 2 animals-11-01676-t002:** Description of the mean differences and correlations of variables in 75 Tibetan sheep (*Ovis aries*) at different stages.

Variable	Month	Mean ± SE	Comparison	Difference		Correlation	
				Z/LSD	*p*	r/rho	*p*
Risk-taking	Mar.	43.37 ± 4.46	Mar.–Jun.	−5.228	<0.001 **	0.183	0.218
	Jun.	14.20 ± 2.73	Jun.–Dec.	−2.500	0.012 *	0.396	0.013 *
	Dec.	7.12 ± 0.84	Mar.–Dec.	−5.223	<0.001 **	−0.013	0.938
Vocalizations	Mar.	23.20 ± 1.83	Mar.–Jun.	−2.075	0.038 *	0.379	0.019 *
	Jun.	19.02 ± 1.62	Jun.–Dec.	−4.376	<0.001 **	0.296	0.151
	Dec.	4.82 ± 0.67	Mar.–Dec.	−4.445	<0.001 **	0.161	0.422
Exploration	Mar.	60.24 ± 4.56	Mar.–Jun.	−0.414	0.679	0.020	0.898
	Jun.	52.32 ± 3.67	Jun.–Dec.	−1.275	0.202	−0.050	0.764
	Dec.	49.95 ± 3.26	Mar.–Dec.	−0.036	0.971	0.354	0.027 *
Novelty	Mar.	68.79 ± 8.15	Mar.–Jun.	−3.404	0.001 **	−0.022	0.903
	Jun.	140.53 ± 6.69	Jun.–Dec.	−0.747	0.455	−0.011	0.959
	Dec.	142.86 ± 8.19	Mar.–Dec.	−3.553	<0.001 **	0.134	0.497
Heart rate	Mar.	237.34 ± 6.12	Mar.–Jun.	−3.045	0.002 **	0.066	0.678
	Jun.	267.55 ± 5.88	Jun.–Dec.	−1.733	0.083	−0.055	0.742
	Dec.	289.71 ± 3.95	Mar.–Dec.	−4.340	<0.001 **	−0.026	0.876
CORT	Mar.	1227.35 ± 56.07	Mar.–Jun.	324.530	0.461	−0.351	0.649
	Jun.	1287.45 ± 60.25	Jun.–Dec.	316.550	<0.001 **	0.710	0.048 *
	Dec.	775.71 ± 51.90	Mar.–Dec.	311.762	<0.001 **	−0.426	0.400

Differences and correlations in CORT were analyzed by one-way ANOVA and the LSD method (LSD0.05) and a Pearson correlation test (r), respectively; differences in the other variables were tested with a non-parametric Wilcoxon signed-rank test (Z) and Spearman correlation test (rho). *p*-values in bold represent significant differences. SE: standard error; LSD: least significant difference. * Difference and correlation are significant at the 0.05 level (two-tailed). ** Difference is significant at the 0.01 level (two-tailed).

## Data Availability

The data presented in this study are available on request from the corresponding author. The data are not yet publicly available due to ongoing analysis by the authors.

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
