# Peer review of "Variations in Behavioral and Physiological Traits in Yearling Tibetan Sheep (Ovis aries)"

_animals, 2021, doi:10.3390/ani11061676_

Round 1

Reviewer 1 Report

General comment

I strongly recommend changing the name of “boldness” and “nervousness” variables. These words are more associated to human’s behavior. For example, “Vocalizations” instead “Nervousness” would be better. I suggest changing both names in the text and tables.

Line 721: The paragraph refers to boldness variable which according to Figure 2a decrease over ontogeny. Please modified this sentence.

Table 2:

I suggest to moving the statistical description (“Differences and correlations in CORT were analyzed……”) at the bottom of the table, added an “*” at the end of the column “Difference and Correlation*”.

Reviewer 2 Report

I agree with all previous corrections that I saw they have accepted and made corrections by the author. 

Author Response

Dear reviewer, many thanks for your comments.

Reviewer 3 Report

Dear authors,

I think you have improved the clarity of your study. The results are interesting and the discussion and conclussions are more balanced  with the results in this new version.

Author Response

Dear reviewer, many thanks for your comments.

This manuscript is a resubmission of an earlier submission. The following is a list of the peer review reports and author responses from that submission.

Round 1

Reviewer 1 Report

There are a number of problems with this manuscript, listed below

  1. The English language and wording/grammar need a major overhaul.
  2. Apart from sentence structure etc, words such as personality, neuroticism etc are associated with anthropomorphism- in this case the word temperament, nervousness, reactivity or nervous temperament would be better.
  3. The assumption is that the correlations between physiological markers and behaviour over time is due to ontogeny - in fact the repeated exposure and conditioning to animal handling by humans will be influencing the reaction of animals (negatively or positively in terms of nervousness indicators). No mention was made of habituation, conditioning or effect of repeated exposure to handling by humans in the manuscript.
  4. Measurement of heart rate by Pratt sound analysis is likely to have a large amount of noise interference and any heart rate measurement will be associated with the level of restraint imposed during variable measurement rather than the effect of a novel object test as the heart rate was not measured remotely. This will not be a valid measurement of reaction to the experimental stress stimuli.
  5. Other comments:
  • Title is confusing with both yearling and different ages- remove last 3 words
  • L12 - change to "are raised in the wild"

It is recommended that the manuscript be rejected in its current form, or largely rewritten in line with the comments provided above.

Reviewer 2 Report

This is a very interesting topic and relevant study. The research is original and is well organized. I have minor comments and changes in order to improve material/methods and result sections.

Simple summary

Line 13: Please improve this sentence (“It is the domesticated for a long period”) in a correct verbal form.

Line 42: Introduction

Include some genetic data related to heritability for personality traits in sheep or similar livestock species (eg. Cattle).

Line 107- 108: Was birth weight measured, and which instrument was used? Please include this information

Line 111-112: Provide information about the selection process (eg., random)

Line 119-121: Was the observer trained and who was the person? Please describe if intra-observer reliability was tested. Moreover, add more details on behavioral recording, including sampling method.

Line 121: Please include “h” in “8:00-13:00”

Line 124-125: How arena was cleaned?

Line 127. I suggest a more detailed title (description) for this table and provide more information for each variable.

Table 1. Neuroticism. Please include units (eg. Frequency - number/minute/hour).

Table 1. Exploration. Please provide more information about this behaviour.

Table 1. Heart rate variable. Please include units.

Line 157: Provide a more detailed description for “body conditions”.

Line 196: The paragraph refers to boldness variable which according to Figure 2a decrease over ontogeny.  Please modified this sentence and make the difference between “more curious - bolder” and boldness variable.

Line 197. According to figure 2d “Novelty” variable increase until 5-month-old.

Line 218. I suggest a more detailed title (description) for this table, including specie, number of animals, experiment.

Reviewer 3 Report

Reviewer(s)’ General Comments to Authors:

The manuscript is overall well written, interesting, and in the aim of the journal. Hence, it can be considered for publication. However, there are some points which need to be improved or changed. I recommend this manuscript for publication in Animals, provided that the minor revision will have been made by the Authors in the re-edited and resubmitted version of the current paper according to the remarks of the Reviewer indicated below:

Line 21, misspelled instead of ‘behavioural’ should be ‘behavioral’ 

Line 22, rephrase word ‘from’ with ‘of’ 

Line 24, Personality is the consistent behavioral differences of individuals over time or IN different contexts.

Line 34, misspelled instead of ‘behavioural’ should be ‘behavioral’

Line 36, recommendation to rephrase word ‘in’ with ‘from’ (... and heart rate FROM the whole period..) 

Line 60, ...over different ontogenic age stages IS still rare..

Link 106-107, write date properly, 4 and 23 of January 2017.

Line 110, misspelled instead of ‘behavioural’ should be ‘behavioral’ 

Line 127, explain a little better (more appropriate) what will be said in table 1 with a description

Line 144, ...while the number of moving grids was used AS the indicator of..

Line 165, it could be helpful to explain this segment a little more

Line 179, difference analysis WAS used to compare

Table 2, full content, to be identical as in the whole text (refers to HR) 

Line 220, Difference and correlation for CORT WERE analyzed.

Line 232-234, The sentences do not convey the message clearly and strongly enough

Line 253, ...following with the growth and development... 

Line 278, ...Bosco et al [25]  reported A strong correlation...

Line 290, rephrase word, ...is independent OF behavior and...

In principle, the study is interesting and comprehensive, however, the manuscript, especially the methods section, is not described enough in some parts. Information on some analysis is missing. Moreover, the conclusion could contain more comments to be more specific and clear for the reader of what your study showed eventually. More detailed information on ethics approval is also needed, so could be a little rephrasing. The reviewer has some minor comments as previously mentioned above before it can be published. 

Reviewer 4 Report

Summary

The aim of this paper it is a better understanding of Tibetan sheep evaluating behavioural (boldness, neuroticism, exploration, novelty) and physiological parameters (heart rate, faecal cortisol) at different ages during their first year of life.

Their main contributions are valuable data regarding changes in boldness (increased with age), neuroticism (decreased), novelty (increased), faecal cortisol decrease in older animals and heart rate increase.

I found this study of interest as there is a lack of studies regarding changes in personality with age in vertebrate species others than humans. However, I have some suggestions,

Broad comments

L35 The results don’t demonstrate that “life experience” affects personality, they show differences in behavioural and physiological parameters at different ages which is not the same. Please reformulate as “age” affects personality.

 Differences could be due to anatomical changes, as for example in the prefrontal cortex, human-animal social experience, intraspecific experience (as you describe in the introduction their contact with ewes and weaning), …You can discuss these questions in the discussion, but not “conclude”.

L120 Use of only one person to analyse de data can be a source of bias and decrease repeatability (external validity). Even if it is a common practice in behavioural studies for logistic reasons and personal resources it should be noted. I suggest including in the discussion this point as one of the weaknesses of the study.

L121 Please describe how did you randomize the capture of the animals. If they were captured without a previous randomization could be a source of bias, as the order of capture could be related to its individual personality and not at random.

L161 Please be more precise about how you selected the peaks of the sound wave

Figure 2 This figure can be misleading in some points at its own and when we read the comments in the results section and in the discussion, so please use the next suggestions:

 There are no units in any of the graphics, they should be explicitly shown in each graphic for each parameter and described with more detail if necessary, in the captions. This problem goes along with discussion (eg L231-232). If you say that boldness increases with age and we see the graphic without any other information as description of units used, seems contradictory, the graphics show decreasing.

The captions should explicitly say which is the descriptive data shown, is the average, the median?

The graphic should be self-explained without need of searching in the material and methods. Please explicit the acronym CORT

Add the statistical methods used in the captions.

Mar Jun Dec don’t mean nothing by themselves animals could have 2 or 3 years in different parts of the year, I suggest writing 3-5-11 months in the graphic for clarity.

Specific comments

L13 I don’t understand “it is the domesticated for a long period”, “the domesticated” seems as the only domesticated animal, and period is not precise. Please be more precise with this sentence if this is what you mean or reformulate if you want to give another sense as “has been domesticated for a long time” or “it has an ancient domestication”

L 16 Please precise 3,5 and 11 months of age for clarity

L18 and 20 The word “varied” it is very imprecise and doesn’t allow to understand the main results of the study, please, precise “increase” or “decrease”, at least with main results.

L33 The same as the previous comment, “varied” doesn’t give any specific information about the results, please precise “increase” or “decrease”.

L238 Reformulate for clarity, seems clearer the order: The personality of wildlife animals may vary with age under harsh…

L257 Consistent with “a previous study”, should be singular, if you use “previous studies” you should cite other studies.

L265 CORT levels indicated that… means causality, which is not proved, I suggest use the form “could indicate” that sheep suffered higher external stress.

L302 funding “by the National” is repeated.
